

# Comparative cytogenetics of three Zoraptera species as a basis for understanding chromosomal evolution in Polyneoptera insects

Marek Jankásek[1], Petr Kočárek[2] and František Šťáhlavský[1]

[1] Department of Zoology, Charles University Prague, Praha 2, Czech Republic
[2] Department of Biology and Ecology, University of Ostrava, Ostrava, Czech Republic

## ABSTRACT

Zoraptera (also called "angel insects") is one of the most unexplored insect orders. However, it holds promise for understanding the evolution of insect karyotypes and genome organization given its status as an early branching group of Polyneoptera and Pterygota (winged insects) during the Paleozoic. Here, we provide karyotype descriptions of three Zorapteran species: *Brazilozoros huxleyi* (2n♂; ♀ = 42; 42), *B. kukalovae* (2n♂; ♀ = 43; 44) and *Latinozoros cacaoensis* (2n♂; ♀ = 36; 36). These species represent two of the four recently recognized Zorapteran subfamilies. Contrary to an earlier suggestion that Zoraptera has holocentric chromosomes, we found karyotypes that were always monocentric. Interestingly, we detected both X0 (*B. kukalovae*) and XY (*B. huxleyi*, *L. cacaoensis*) sex chromosome systems. In addition to conventional karyotype descriptions, we applied fluorescent *in situ* hybridization for the first time in Zoraptera to map karyotype distributions of 18S rDNA, histone H3 genes, telomeres and $(CAG)_n$ and $(GATA)_n$ microsatellites. This study provides a foundation for cytogenetic research in Zoraptera.

## INTRODUCTION

Zoraptera is the third smallest insect order and also one of the most enigmatic (*Choe, 2018*; *Matsumura et al., 2020*). It originated during the Paleozoic (*Matsumura et al., 2020*; *Misof et al., 2014*; *Montagna et al., 2019*; *Wang et al., 2023*), and the majority of its extant species are found living under the bark of fallen logs in subtropical and tropical forests in all biogeographical regions (reviewed in *Choe, 2018*). Since being described by *Silvestri (1913)* more than a century ago, Zoraptera has received scant attention. Knowledge of the biology of this insect order is therefore very limited. Moreover, locating Zoraptera in the insect phylogeny tree has been shown to be a difficult task—labelling it the "Zoraptera problem" (*Beutel & Weide, 2005*). The group was either proposed to be sister to Paraneoptera (also called Acercaria) (*Beutel & Weide, 2005*; *Hennig, 1953*; *Wheeler et al., 2001*) or Eumetabola (Paraneoptera + Holometabola) (*Beutel & Gorb, 2001*; *Blanke et al., 2012*) or to be an inner lineage of Polyneoptera (reviewed by *Choe, 2018*). Even within Polyneoptera, the

Corresponding author
František Šťáhlavský,
frantisek.stahlavsky@natur.cuni.cz

position of Zoraptera was not stable. *Silvestri (1913)* originally assumed its close affinity to Blattodea (cockroaches and termites). More recent phylogenetic studies have proposed sister relationships to Dictyoptera (Blattodea + Mantodea) (*Ishiwata et al., 2011*; *Wang et al., 2013*; *Wheeler et al., 2001*; *Yoshizawa & Johnson, 2005*), Embioptera or the entire Eukinolabia (Embioptera + Phasmatodea) (*Dallai et al., 2011*; *Dallai et al., 2012*; *Ma et al., 2014*; *Mashimo et al., 2011*; *Mashimo et al., 2015*) or Dermaptera (*Misof et al., 2014*; *Misof et al., 2007*; *Wipfler et al., 2019*). Recently, there seems to be consensus that Zoraptera represents an early branching lineage of Polyneoptera; phylogenomic studies published in the last few years have supported its sister position to Dermaptera (*Misof et al., 2014*; *Wipfler et al., 2019*) or to all other Polyneoptera (*Tihelka et al., 2021*). Only 61 Zorapteran species have been described to date. However, considerable systematic effort has been recently focused on inner classification of the group, suggesting that its true diversity remains largely unexplored (*Matsumura et al., 2020*; *Kočárek, Horká & Kundrata, 2020*; *Kočárek & Horká, 2023a*; *Kočárek & Horká, 2023b*; *Matsumura et al., 2023*). The group is classified into families Spiralizoridae and Zorotypidae. Spiralizoridae is further divided into subfamilies Latinozorinae and Spiralizorinae and Zorotypidae is divided into subfamilies Spermozorinae and Zorotypinae (*Kočárek, Horká & Kundrata, 2020*).

The phylogenetic position of Zoraptera makes the group important for understanding the evolution of insect karyotype characteristics—including sex chromosome systems (SCSs), chromosome number and holocentricity. However, even basic descriptions of Zorapteran karyotypes are still not available. The only exception is in case of *Usazoros hubbardi* (Caudell, 1918), whose male karyotype has been shown to constitute 38 chromosomes ($2n\male = 38$) (*Kuznetsova, Nokkala & Shcherbakov, 2002*). Those same authors also suggested that an XY SCS is present in this species and that its chromosomes are holocentric because no primary constriction was observed. Among insects, holocentric chromosomes are also found in Odonata, Dermaptera, Phthiraptera, Hemiptera, Trichoptera and Lepidoptera (reviewed in *Drinnenberg et al., 2014*). The phylogenetic positions of these lineages (including Zoraptera) support the idea that holocentric chromosomes may be ancestral for Pterygota according to maximum parsimony analysis performed by *Melters et al. (2012)*. However, the transcriptomic and genomic analyses of *Drinnenberg et al. (2014)* revealed that Odonata, Dermaptera, Phthiraptera, Hemiptera, Trichoptera and Lepidoptera lack CENP-A (centromeric histone 3 variant), which is otherwise essential for inner kinetochore construction in eukaryotes, including the insect groups with monocentric chromosomes. *Drinnenberg et al. (2014)* concluded that the loss of CENP-A and the establishment of holocentricity occurred several times independently in insects; that finding is at odds with the ancestral holocentricity hypothesis (*Melters et al., 2012*). Importantly, those authors also suggested that perception of some insect orders to have exclusively holocentric chromosomes might be oversimplified because of limitations related to the karyological data (*Drinnenberg et al., 2014*); such limitations could bias large-scale evolutionary metanalyses.

In order to broaden our knowledge on Zorapteran cytogenetics, we provide karyotype descriptions of three species belonging to the sister subfamilies Latinozorinae (*Latinozoros*) and Spiralizorinae (*Brazilozoros*) (*Kočárek, Horká & Kundrata, 2020*). In addition to

standard cytogenetic techniques, we also performed fluorescent *in situ* hybridization (FISH) to map and compare the karyotype distribution of five different repetitive DNA markers. Generally, karyotype distribution of various repetitive DNA loci is frequently mapped in cytogenetic analyses to study their role in chromosomal rearrangements and its spatial organization and function in genomes (*e.g.,* *Cazaux et al., 2011*; *Raskina et al., 2008*; *Slijepcevic, 1998*; *Štundlová et al., 2022*; *Voleníková et al., 2023*). In this study, we analyzed distribution of telomeric repeats, 18S rDNA and histone H3 gene, which are the most frequently studied cytogenetic markers in animals and plants (*e.g.,* *Fuková, Nguyen & Marec, 2005*; *Roa & Guerra, 2012*; *Rovatsos et al., 2015*; *Šťáhlavský et al., 2021*; *Šťáhlavský et al., 2020*; *Šťáhlavský et al., 2018*). On the other hand, microsatellite sequences constitute a significant fraction of the repeatomes of eukaryotic organisms (*Vieira et al., 2016*) and may mediate centromere and telomere formation, gene expression regulation, chromatin organization and DNA structure (reviewed in *Jonika, Lo & Blackmon 2020*). However, studies focusing on the spatial organization of microsatellite sequences within genomes have been rare; only a few investigations have involved insects (*e.g.,* *Milani & Cabral-de Mello, 2014*; *Palacios-Gimenez et al., 2015a*; *Palacios-Gimenez, Marti & Cabral-de Mello, 2015b*; *Palacios-Gimenez & Cabral-de Mello, 2015*; *Panzera et al., 2023*; *Ruiz-Ruano et al., 2015*; *Santos et al., 2010*; *Dos Santos et al., 2018*). In this study, we analyze the distribution of $(GATA)_n$ and $(CAG)_n$ microsatellite sequences in Zoraptera. These two types of microsatellites have been reported to be present in plants (*e.g.,* *Stajner et al., 2005*; *Vosman & Arens, 1997*), vertebrates (*e.g.,* *Andrés et al., 2004*; *Haerter et al., 2023*; *Hiramatsu et al., 2017*; *Liang et al., 2007*; *Lin, Dion & Wilson, 2005*; *Mubiru et al., 2012*; *Tokarskaya et al., 2004*) and insects (*e.g.,* *Milani & Cabral-de Mello, 2014*; *Palacios-Gimenez et al., 2015a*; *Palacios-Gimenez, Marti & Cabral-de Mello, 2015b*; *Ruiz-Ruano et al., 2015*), which suggests that their presence might be widespread in eukaryotic organisms and represent suitable markers to study the differences of chromosome organization among even closely related species (*e.g.,* *Palacios-Gimenez et al., 2015a*; *Pucci et al., 2016*).

## METHODS

### Sampling, rearing, and identification

Zorapteran specimens used in this study were collected during the expedition to French Guiana in 2022 by P. Kočárek, M. Jankásek (authors of this study), and I.H. Tuf (Palacký University, Olomouc, Czech Republic). An aspirator was used to collect zorapteran specimens from under the bark of different tree species. Breeding cultures were established from the collected individuals and the zorapterans were reared in plastic containers ($12.0 \times 9.5 \times 4.5$ cm) in crushed oak rotting wood at room temperature (22–24 °C). The samples for chromosome slide preparations (see below) were extracted from these cultures. Breeding cultures of *Latinozoros cacaoensis* Kočárek & Horká, 2023 and *Brazilozoros kukalovae* Kočárek & Horká, 2023 have been established from samples collected at the type localities together with the type material (*Kočárek & Horká, 2023*; *Kočárek & Horká, 2023b*). *Brazilozoros huxleyi* (Bolívar y Pieltain & Coronado, 1963) has been identified morphologically by comparisons with all described species of *Brazilozoros* Kukalova-Peck

& Peck, 1993 (*Silvestri, 1946*; *Bolívar Y Pieltain & Coronado, 1963*; *New, 1978*; *Kočárek & Horká, 2023b*). Phylogenetic affinities of studied species have been compared molecularly (*Kočárek & Horká, 2023a*; *Kočárek & Horká, 2023b*).

We analyzed karyotypes of three Zorapteran species in the family Spiralizoridae from French Guiana: *Latinozoros cacaoensis* (Latinozorinae), eight males, four females, locality: Cacao env., Molokoï track (04°33′39.70″N, 52°27′44.52″W); *Brazilozoros huxleyi* (Spiralizorinae), five males, three females, locality: Kourou env., Montagne des Singes (05°04′17.11″N, 52°41′50.26″W); *Brazilozoros kukalovae* (Spiralizorinae), seven males, seven females, locality: Kourou env., Montagne des Singes (05°04′17.11″N, 52°41′50.26″W). Voucher specimens are deposited in the Department of Zoology, Charles University in Prague.

## Chromosome slide preparations

The chromosome preparation slides were prepared *via* the "plate spreading" method following *Traut (1976)*. The abdomen cavity of an individual was opened and hypotonized in 0.075 M KCl solution for 35 min. Next, the entire abdomen was fixed in a methanol:acetic acid (3:1) solution for 20 min and its content was dissolved on a slide in a drop of 60% acetic acid. Subsequently, the suspension was dried and spread on a histological plate at 45 °C and the chromosomes were stained in 5% Giemsa-Romanowski solution in Sörensen phosphate buffer.

## Microsatellite fluorescent *in situ* hybridization experiments

Biotin-labelled $(GATA)_8$, $(CAG)_{10}$ probes and Cy3-labelled $(TTAGG)_8$ probes (Integrated DNA Technologies, Inc., Coralville, USA) were used to detect the distributions of microsatellite and telomeric sequences in all the studied species. In order to test specificity of the telomeric $(TTAGG)_8$ probe, we also used $(TTAGGG)_8$ probe in *L. cacaoensis.* The FISH experiments were performed following a modified version of the non-denaturing protocol presented by *Cuadrado & Jouve (2010)*. Briefly, no pre-treatment of slides has been performed and 30 μl of hybridization buffer consisting of 2 pmol of probe in 2xSSC was denatured in 80 °C for 5 min, chilled on ice for 10 min and applied to each slide. The hybridization process was conducted over the course of 2 h at room temperature, and the slides were washed for 10 and 5 min in 4xSSC/0.2% Tween afterwards. For the biotin-labelled probes, 100 μl of detection mix consisting of streptavidin-Cy3 in 5% BSA/4xSSC (2:1000) was applied to each slide, and the slides were incubated for 1 h at 37 °C before being washed for 10 and 5 min in 4xSSC/0.1% Tween. Finally, all of the slides were stained using commercial Mounting Medium with DAPI (Abcam plc., Cambridge, UK).

## 18S rDNA and H3 probe preparation

The 18S rDNA probe (~1,200 bp) was amplified *via* PCR from genomic DNA of *B. huxleyi* (GenBank: PQ315823, File S2) using the primers eukA: 5′-AACCTGGTTGATCCTGCCAGT-3′ and eukB: 5′-TGATCCTTCTGCAGGTTCACCTACG-3′(*Medeli et al., 1988*) under the following conditions: 95 °C for 3 min, 35 cycles of 95 °C for 30 s, 51 °C for 40 s and 72 °C for 2 min. The final extension was at 72 °C for 10 min. The PCR product was purified

using a Gel/PCR DNA Fragments Kit (Geneaid Biotech Ltd., New Taipei City, Taiwan) and labelled using a Cy3 NT Labeling Kit (Jena Bioscience, Jena, Germany) following the manufacturer's protocols. Next, the solution containing the labelled probe and competitive Salmon Sperm DNA was ethanol precipitated. The final hybridization mixture contained 20 ng of the probe and 25 µg of competitive Salmon Sperm DNA per slide diluted in 50% formamide/2xSSC (5.2 µl/slide) and 10% dextran sulphate (5.2 µl/slide) at 37 °C.

The probe for histone 3 (H3) gene detection was prepared specifically for *B. huxleyi* (GenBank: PQ309679, File S2) and *L. cacaoensis* (GenBank: PQ309678, File S2). It was amplified and labelled with biotin-16-dUTP *via* PCR using the following primers: H3 AF: 5′-ATGGCTCGTACCAAGCAGACVGC-3′, H3 AR: 5′-ATATCCTTRGGCATRATRGTGAC-3′(*Colgan Donald, Ponder & Eggler, 2000*). The PCR conditions were as follows: 95 °C for 5 min, 35 cycles of 95 °C for 1 min, 52 °C for 1 min and 72 °C for 1 min 20 s. The final extension was at 72 °C for 7 min. After being mixed with competitive Salmon Sperm DNA, the probe solution was ethanol precipitated; the final hybridization mixture contained 100 ng of the probe and 25 µg of competitive Salmon Sperm DNA per slide diluted in 50% formamide/2xSSC (5.2 µl/slide) and 10% dextran sulphate (5.2 µl/slide) at 37 °C.

### H3 and 18S rDNA FISH

The histone H3 and 18S rDNA FISH experiments were conducted following a modified version of the protocol of *Fuková, Nguyen & Marec (2005)*. Briefly, the slides were pre-treated with 100 µl of RNase A (200 µg/ml in 2xSSC) for 60 min, washed twice in 2xSSC for 5 min and denatured in 70% formamide for 3 min 30 s. Next, the hybridization mixtures were applied, and the hybridization process took place for 16–22 h. The stringency washes were performed following *Sahara, Marec & Traut (1999)*. The slides with the H3 biotin-16-dUTP-labelled probe were incubated with 500 µl of 2.5% BSA /4xSSC blocking reagent for 20 min. Detection of the probe was carried out using 100 µl of streptavidin-Cy3 in 2.5% BSA/4xSSC (1:1000) for 1 h and was followed by 3 rounds of washing in 4xSSC/0.2% Tween for 3 min. Finally, all of the slides were counterstained using Mounting Medium with DAPI (Abcam plc., Cambridge, UK) The slides selected for sequential reprobing after FISH were washed according to *Štundlová et al. (2022)*.

### Microscopy and imaging

The chromosomes subjected to Giemsa-Romanowski staining and the FISH experiments were photographed using an Olympus AX70 Provis microscope with an Olympus DP72 camera and fluorescent filters. The chromosomes of five mitotic metaphases were measured in ImageJ 1.53p (*Schneider, Rasband & Eliceiri, 2012*) using the Levan plugin (*Sakamoto & Zacaro, 2009*) and categorized into morphological categories according to *Levan, Fredga & Sandberg (1964)*.

## RESULTS

### *Brazilozoros huxleyi* (Bolívar y Pieltain & Coronado, 1963)

This species exhibited 2n = 42 in female mitotic metaphases (Figs. 1A, 1B). The karyotype was composed of one pair of metacentric (pair No. 1) and 19 pairs of telocentric

autosomes. Moreover, females also possessed two large metacentric sex X chromosomes. The metacentric autosomes and the X chromosomes constituted the largest chromosomes in the karyotype; each of them accounted for 5.65 and 5.78%, respectively, of the diploid set. The remaining autosomes gradually decreased in length from 3.06 to 1.26% of the diploid set. The males of this species possessed 20 homomorphic and one heteromorphic chromosome pair (Figs. 1E, 1F). The heteromorphic chromosome pair represented the XY SCS. The *Y* chromosome was distinctly shorter; it constituted 63.4% (SE = 5.5) of the *X* chromosome. These sex chromosomes exhibited intense spiralization that manifested itself as positive heteropycnosis during the first meiotic division (pachytene and diplotene) (Figs. 1E, S1A). The application of FISH with the $(TTAGG)_8$ probe allowed us to detect typical telomeric signals in the terminal regions of all of the chromosomes and moreover two small pericentric signals on the large metacentric pair No. 1 (Figs. 1B, 1C). The position of the histone H3 gene was terminal on the proposed short arms of one chromosome pair (Fig. 1D); the $(CAG)_n$ microsatellite clusters were pericentric on the long arms of another chromosome pair (Fig. 1D). Unlike previously reported markers, the 18S rDNA cluster was identified on a different chromosome pair in the interstitial position of the long arms, app. in the middle of their length (Fig. 1D). The $(GATA)_8$ probe yielded a specific chromosomal pattern. Except for a pair of very intense pericentric signals on one autosome, we also identified weak signals in the short arm (peri)centromeric regions of all of the autosomes. Interestingly we also identified small signal in terminal position of the proposed metacentric X chromosome (Fig. 1F).

### *Brazilozoros kukalovae* Kočárek & Horká, 2023

This species exhibited 2n = 43 in the male mitotic metaphases (Figs. 2A, 2B) and 2n = 44 in the female mitotic metaphases (Figs. 2D, S1D). A comparison of the males and females revealed the presence of an X0 SCS in this species. This system could also be identified from the male metaphases II. One half of the observed metaphases II contained 21 chromosomes (Fig. S1B); the remaining metaphases contained 22 chromosomes with a slightly positive heteropycnotic X chromosome (Fig. 2C). The karyotype was composed only of telocentric chromosomes (including the X). The X chromosome is the largest in the karyotype; it represents 3.99% of the diploid set length. The remaining autosomes gradually decrease in length from 2.96 to 1.38% of the diploid set. The application of FISH with the $(TTAGG)_8$ probe enabled the detection of typical telomeric signals in the terminal regions of all of the chromosomes (Fig. 2B). Moreover, we identified one distinct interstitial signal on chromosome pair No. 2 and No. 6 (Fig. 2B). We detected $(CAG)_n$ microsatellite clusters on the long arm of one chromosome pair (Fig. 2D). Clusters of 18S rDNA and $(GATA)_n$ microsatellites were additionally identified on another chromosome pair in an interstitial position of the long arms (Figs. 2E, 2F). We did not detect any additional accumulations of (GATA) repeat units.

### *Latinozoros cacaoensis* Kočárek & Horká, 2023

This species exhibited 36 chromosomes in the female mitotic metaphases (Figs. 3A, 3B). During diakinesis and metaphase I, the males possessed 17 pairs of autosomes and X

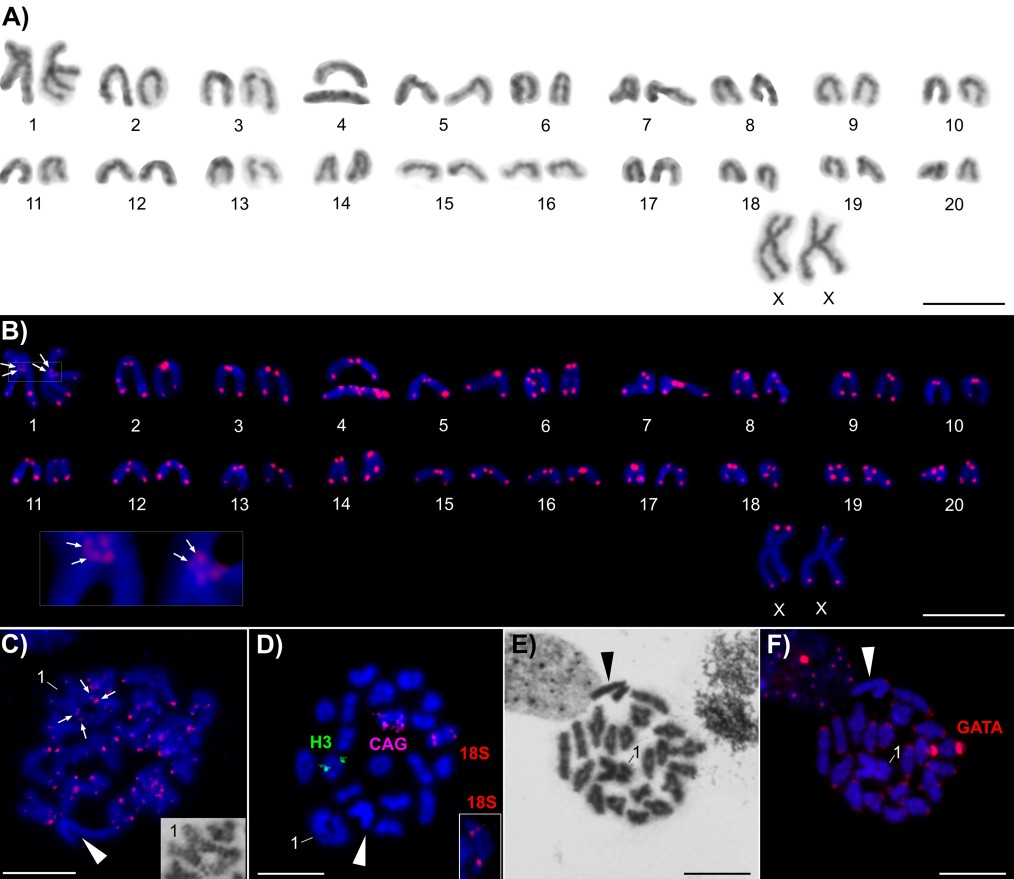

**Figure 1  Chromosomes of *Brazilozoros huxleyi*.** The chromosomes are counterstained with Giemsa (A, E) or with DAPI (blue) (B–D, F). The arrows indicate the small pericentric signals; the white arrowheads indicate the sex chromosomes; "1" indicates bivalent of chromosome pair No. 1. Each scale bar corresponds to 10 μm. (A) Female karyotype (based on mitotic metaphase). (B) Female karyotype after FISH with the (TTAGG)$_8$ probe (based on the same mitotic metaphase appearing in the previous karyogram); the inset shows details of the centromeric region of pair No. 1. Two small pericentric signals are noted with the telomeric probe. (C) Male diplotene after FISH with the (TTAGG)$_8$ probe; the inset shows the same bivalent pair No. 1 after Giemsa staining. (D) Male diakinesis after FISH with the histone H3 gene probe (green signals), the (CAG)$_{10}$ probe (magenta signals) and the 18S rDNA probe (red signals). The inset shows details from another diplotene. (E) Male diplotene with slightly positive heteropycnotic sex chromosomes. (F) The same male diplotene after FISH with the (GATA)$_8$ probe (red signals).

and Y sex chromosome univalents (Fig. 3C). The first two pairs of chromosomes were distinctly longer than the remaining chromosomes—they were metacentric and accounted for 6.27% and 3.97%, respectively, of the diploid set length. Moreover, both of these long autosomal pairs formed two chiasmata during the first meiotic division (Fig. 3C). The X chromosome also represents a relatively large metacentric chromosome that accounted for 5.45% of the diploid set length; the Y chromosome is a telocentric chromosome that was only about 40% of the length of the X chromosome. Moreover, the sex chromosomes appeared to be positively heteropycnotic during early prophase I. All of the autosomes gradually decreased in length and accounted for anywhere from 2.92% to 1.36% of the

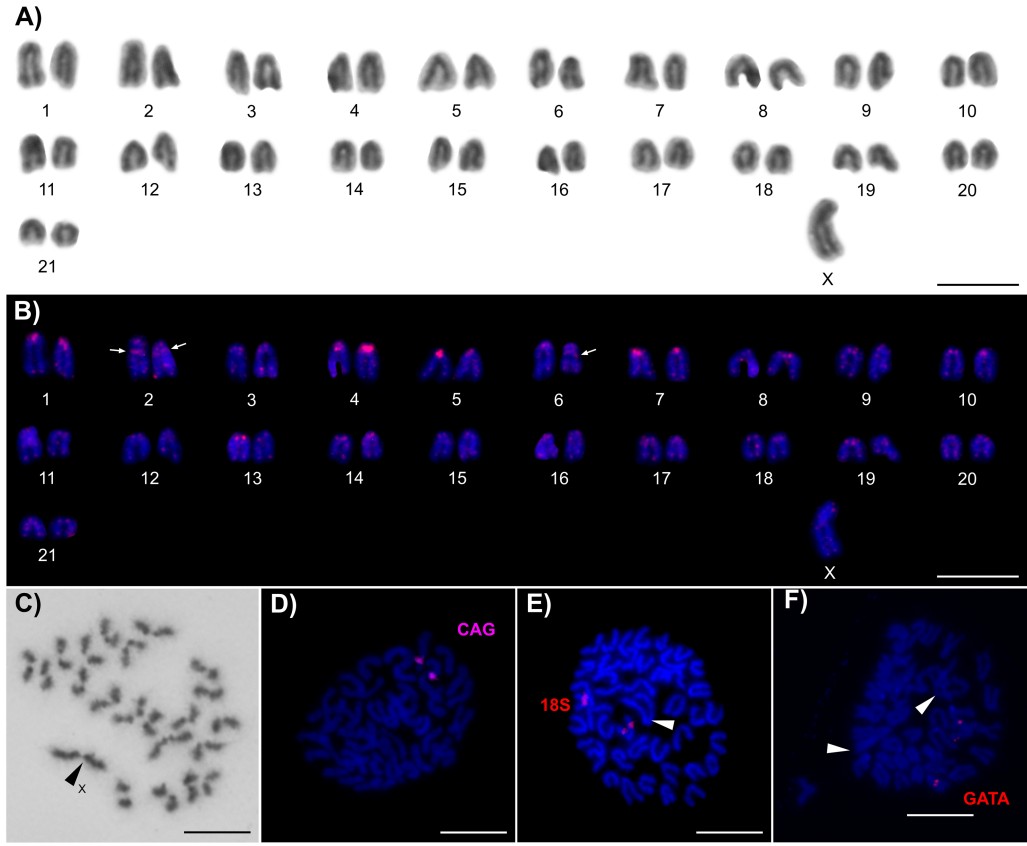

**Figure 2  Chromosomes of *Brazilozoros kukalovae*.** The chromosomes are counterstained with Giemsa (A, C) or with DAPI (blue) (B, D–F); the arrowheads point out sex chromosomes. Each scale bar corresponds to 10 µm. (A) Male karyotype (based on mitotic metaphase). (B) Male karyotype after FISH with the (TTAGG)$_8$ probe (based on the same mitotic metaphase appearing in the previous karyogram); the arrows indicate small interstitial signals. (C) One sister cell of male metaphase II with the X chromosome. (D) Female mitotic metaphase after FISH with the (CAG)$_{10}$ probe (magenta signals). (E) Male mitotic metaphase after FISH with a probe for 18S rDNA (red signals). (F) Female mitotic metaphase after FISH with the (GATA)$_8$ probe (red signals).

diploid set length. These autosomes were mainly telocentric with the exception of two pairs of submetacetrics and two pairs of subtelocentrics. FISH with the (TTAGG)$_8$ and (TTAGGG)$_8$ probes revealed standard telomere patterns on all of the chromosomes (Figs. 3B, S1C). To further test the specificity of both probes, we also applied stronger stringency washes following the methodology in the H3 and 18S rDNA FISH experiments. However, this approach did not provide any reliable signals. Clusters of histone H3 genes and (CAG) microsatellite repeats were detected in the pericentric regions of the short and long arms of two different telocentric chromosomes, respectively (compare Figs. 3D, 3E). Finally, one pair of 18S rDNA clusters was detected throughout the entire short arms of a submetacentric chromosome pair (Fig. 3E). Unlike previous markers, we identified multiple smaller (GATA)$_n$ microsatellite clusters on six chromosomal pairs and three in a heterozygous state on two chromosomal pairs (Fig. 3F).

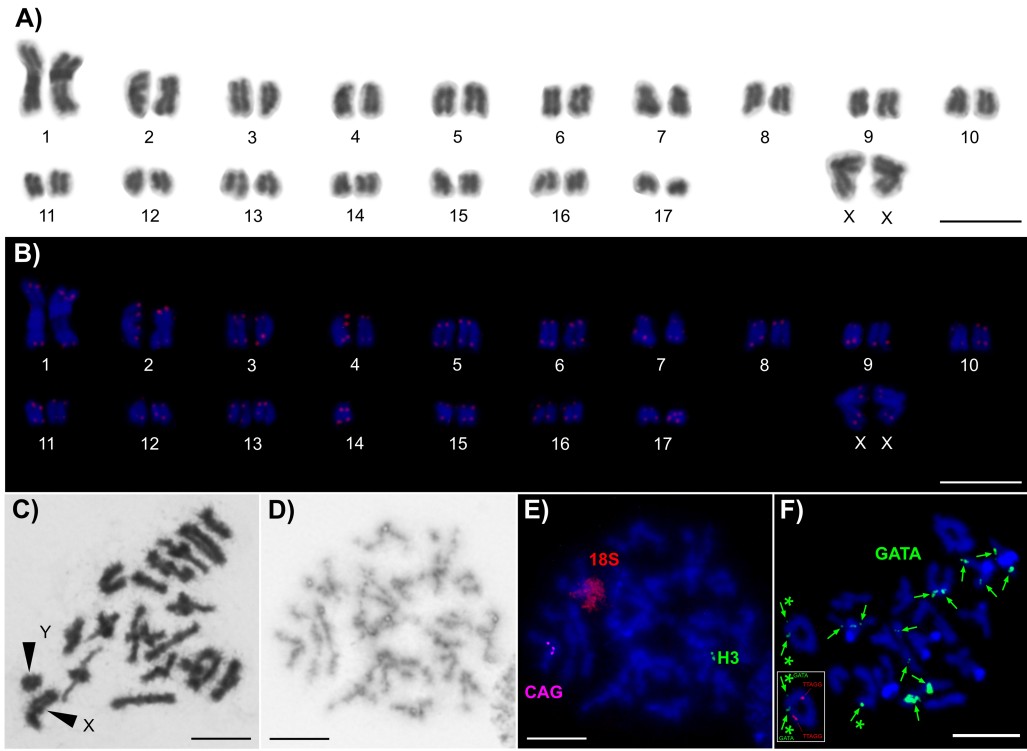

**Figure 3** **Chromosomes of *Latinozoros cacaoensis*.** The chromosomes are counterstained with Giemsa (A, C, D) or with DAPI (blue) (B, E, F). The scale bars correspond to 10 μm. (A) Female karyotype (based on mitotic metaphase). (B) Female karyotype after FISH with the $(TTAGG)_8$ probe (based on the same mitotic metaphase appearing in the previous karyogram). (C) Male metaphase I; the arrowheads point out sex chromosomes. (D) One male sister metaphase II. (E) The same metaphase II as in the previous panel after FISH with the histone H3 gene probe (green signals), the $(CAG)_{10}$ probe (magenta signals) and the probe for 18S rDNA (red signals). (F) Male diplotene after FISH with the $(GATA)_8$ probe (pointed by green arrows, the asterisks show heterozygous signals); the inset shows the bivalent with two heterozygous signals on one chromosome (red signal: $(TTAGG)_n$).

## DISCUSSION

We have presented detailed cytogenetic analyses of three Zoraptera species representing two subfamilies belonging to the family Spiralizoridae. The observed diploid chromosome numbers in *B. huxleyi*, *B. kukalovae* and *L. cacaoensis* were 2n♂, ♀ = 42, 42; 2n♂, ♀ = 43, 44 and 2n♂, ♀ = 36, 36, respectively. The number of diploid chromosomes in Polyneoptera ranges from 7 in males of *Hemimerus bouvieri* (Hemimeridae, Dermaptera) (*White, 1971*) to 98 in both sexes of *Mastotermes darwiniensis* (Mastotermitidae, Blattodea) (*Bedo, 1987*; *Luykx, 1990*). According to the Polyneoptera karyotype database assembled by *Sylvester et al. (2020)*, the modal chromosome numbers of Polyneoptera are 2n♂, ♀ = 23, 24. However, these numbers are not very informative in the context of the entire Polyneoptera group, because it is prevalent in Caelifera, an extensively karyotyped group (*Husemann et al., 2022*; *Sylvester et al., 2020*). Moreover, *Sylvester et al. (2020)* showed that relatively more cytogenetically studied Polyneoptera orders (Blattodea, Mantodea, Orthoptera and Phasmatodea) have divergent karyotype evolution modes (*i.e.,* differing optimal rates of

chromosome fission, fusion and polyploidy). However, the karyotype evolution modes of cytogenetically less studied Polyneoptera orders remain unknown. The karyotypic descriptions of *B. huxleyi*, *B. kukalovae* and *L. cacaoensis* presented here and those of *U. hubbardi* presented by *Kuznetsova, Nokkala & Shcherbakov (2002)* represent an initial body of knowledge for karyotype evolution studies in Zoraptera. *Kuznetsova, Nokkala & Shcherbakov (2002)* suggested that the chromosomes of *U. hubbardi* are holocentric, but the entire chromosome complements of the species we studied are clearly monocentric. However, variability in kinetochore organization (monocentricity or holocentricity) cannot be ruled out in Zoraptera, since *U. hubbardi* represents family Zorotypidae and all of the species sampled here belong to sister family Spiralizoridae. This needs to be confirmed with additional analyses since the chromosomal images presented by *Kuznetsova, Nokkala & Shcherbakov (2002)* are not of sufficient quality to address this issue. Furthermore, it has been proposed that holocentricity might possibly represent synapomorphy for the Dermaptera + Zoraptera group (*Kuznetsova, Nokkala & Shcherbakov, 2002*), but that seems to be uncertain in light of our results and recent phylogenomic research in Polyneoptera, where Zoraptera was found to be sister to all other Polyneoptera (*Tihelka et al., 2021*).

The differing chromosome numbers between *U. hubbardi* and the species studied herein suggest that Zoraptera are not in a karyotype evolution stasis, a phenomenon which is sometimes present even in specious insect groups (*e.g.* in termites of family Termitidae (reviewed in *Jankásek, Kotyková Varadínová & Šťáhlavský, 2021*) or Acrididae grasshoppers (*Husemann et al., 2022*)). All three of the species that we examined possessed karyotypes composed entirely or mostly of (sub)telocentric chromosomes. These chromosomes may undergo Robertsonian fusions (also frequently called centric fusions or Robertsonian translocations). The presence of two large metacentric chromosomes in *B. huxleyi* (2n♂ = 40 + XY) and their lack in *B. kukalovae* (2n♂ = 42 + X0) might suggest that the two metacentric chromosomes originated by fusions. However, exact directionality of the process cannot be revealed without reconstruction of the ancestral *Brazilozoros* karyotype. Therefore, future studies focused on inter-specific applications of specific chromosome painting probes and/or comparative chromosome-scale macrosynteny analyses are necessary to better understand the cross-species relationships between particular chromosomes.

*Kuznetsova, Nokkala & Shcherbakov (2002)* suggested the presence of an XY SCS in *U. hubbardi* since they observed only bivalents in the first meiotic divisions in males. We detected an XY SCS in *B. huxleyi* and *L. cacaoensis* and an X0 SCS in *B. kukalovae*. It is tempting to imagine that an XY SCS is ancestral in Zoraptera since it has been found in three genera (including *Usazoros*) characterized by deep evolutionary divergencies (*Kočárek, Horká & Kundrata, 2020*; *Matsumura et al., 2020*) and could be shared with Dermaptera, which have been considered to be sister to Zoraptera (*Wipfler et al., 2019*) (but see *Tihelka et al., 2021*), where XY (or XY derived) SCS is as far as known predominant (*Blackmon, Ross & Bachtrog, 2017*; *White, 1976*). In that case, it might be possible that the X0 SCS in *B. kukalovae* was derived from an XY SCS *via* the degradation and loss of a Y chromosome. However, the homeology of the X and Y chromosomes across Zoraptera needs to be tested, particularly since it has been shown that the emergence of neo-XY SCSs

from an X0 SCS *via* X chromosome-autosome fusions is frequent in insects (*Blackmon, Ross & Bachtrog, 2017*). In support of the alternative hypothesis of an ancestral X0 SCS, the X chromosome of *B. kukalovae* differed from the metacentric X chromosomes of the other two studied species in that it was telocentric. That situation makes it easy to form neo-XY SCSs *via* Robertsonian fusions. Nevertheless, the homeology of the observed X chromosomes still needs to be confirmed, even though it has been shown recently that the ancestral X chromosome genetic linkage group is conserved in X chromosomes across Hexapoda with some exceptions in Diptera and in Lepidoptera where sex chromosome turnovers occurred (*Li, Mank & Ban, 2022*; *Meisel, Delclos & Wexler, 2019*; *Toups & Vicoso, 2023*).

Only one pair of 18S rDNA clusters was detected in all three of the Zorapteran species that we studied. In animals and plants, rDNA clusters are most frequently found terminally and/or on short arms of (sub)telocentric chromosomes (*Roa & Guerra, 2012*; *Roa & Guerra, 2015*; *Sochorová et al., 2018*). On the other hand, the interstitial position of 18S rDNA on a long chromosomal arm, which was observed in both studied *Brazilozoros* species, is the least common type of rDNA localization in animals and plants (*Roa & Guerra, 2012*; *Roa & Guerra, 2015*; *Sochorová et al., 2018*). Even so, it is more frequent in arthropods than in other animals (*Sochorová et al., 2018*). The low incidence of interstitial rDNA clusters (as observed in both of the *Brazilozoros* species we studied) is not fully understood; however, the more-prevalent distribution of rDNA in terminal chromosome regions compared with elsewhere might be due to the spatial organization of chromosomes into bouquet formation during meiotic prophase. The bouquet formation is present in many animals from the leptotene stage to the pachytene stage; during that time, the telomeres are clustered in one region of the inner nuclear membrane causing the chromosomes to form loops into the nucleus. The proximity of the terminal regions of non-homologous chromosomes in this stage increases the likelihood of ectopic recombination (*Goldman & Lichten, 1996*; *Goldman & Lichten, 2000*; *Penfold et al., 2012*). As a result, even rDNA might be transposed in these terminal regions and therefore be more frequent there than in interstitial regions (*Cazaux et al., 2011*).

Unlike rDNA, the number of clusters containing histone H3 genes appears to be conservative among insects; most of the groups studied thus far have had at most one cluster per haploid set, *e.g.,* Cicadomorpha (*Anjos et al., 2019*), Scarabaeidae beetles (*Cabral-de Mello, Moura & Martins, 2010*; *Cabral-de Mello, Moura & Martins, 2011*; *Cabral-De-Mello et al., 2011a*; *Cabral-de Mello et al., 2011b*); grasshoppers (*Cabral-De-Mello et al., 2011a*; *Cabrero et al., 2009*; *Camacho et al., 2015*). The same situation was also observed in our study of *B. huxleyi* and *L. cacaoensis*. The localization and transposition of histone genes is supposedly subject to the same mechanisms as in case of rDNA and other repetitive DNA sequences (*Bueno, Palacios-Gimenez & Cabral-de Mello, 2013*). Individual histone genes (H4, H3, H2A, H2B and H1) are generally arranged in numerous tandem repeats, which enables their detection *via* standard FISH techniques (*e.g.,* *Cabral-De-Mello et al., 2011a*; *Cabrero et al., 2009*). However, individual clusters may differ in their composition of histone genes and their sequence order within individual repeats, *i.e.,* different histone gene combinations may be present in different clusters in various organisms (reviewed in

*Maxson et al. 1983*). Thus far, the colocalization of H3 and H4 genes has been confirmed in Acridid grasshoppers (*Cabrero et al., 2009*), and the presence of all of the histone genes (H2a, H2b, H3, H4 and H1) in a single locus has been confirmed in *Drosophila melanogaster* (reviewed in *Koreski et al., 2020*). Therefore, it is of interest to test the colocalization of histone genes in Zoraptera as well as in other insect species in the future.

In all three of the Zorapteran species we studied, $(CAG)_n$ microsatellite clusters were always detected in the pericentric region of one telocentric chromosome pair. This finding suggests that these chromosomes are homeologous and that the $(CAG)_n$ band serves as their marker. Similarly, $(CAG)_n$ accumulations have been detected in the centromeres of all three of the autosomal chromosome pairs and the $X_1$ and $X_2$ chromosomes of the *Eneoptera surinamensis* cricket (*Palacios-Gimenez et al., 2015a*). Among Polyneoptera, the $(CAG)_n$ microsatellite has been also detected in low abundances in genomes of *Eyprepocnemis plorans* and *Locusta migratoria* grasshoppers without forming any FISH detectable clusters (*Ruiz-Ruano et al., 2015*).

The distribution of GATA microsatellite accumulations was largely distinct in each of the species we studied. Moreover, a heterozygous state—in sense of presence/absence of detectable GATA clusters—was observed on two autosomal chromosome pairs in *L. cacaoensis*. Among other insects, distribution patterns of GATA microsatellite have been mostly studied in Orthoptera. The findings are generally congruent with our results and show largely differing and specific GATA repeats distributions (*Palacios-Gimenez & Cabral-de Mello, 2015*; *Palacios-Gimenez & Cabral-de Mello, 2015*; *Ruiz-Ruano et al., 2015*). However, these studies always compared karyotypes of different genera and did not focus on differences between closely related species. Contrary to that, the GATA repeats were found to be almost exclusively on Y chromosome across closely related species of Triatomini (Reduviidae, Hemiptera) (*Panzera et al., 2023*). The GATA microsatellite has been also found to be frequently located on sex chromosomes of various vertebrates (*Arnemann et al., 1986*; *Jones & Singh, 1981*; *Nanda et al., 1990*; *Schäfer et al., 1986*; *Singh et al., 1994*; *Subramanian, Mishra & Singh, 2003*), where it has been shown to downregulate or upregulate expression *via* GATA-binding proteins (*Ravid Lustig et al., 2023*; *Singh et al., 1994*). Concerning the sex chromosomes, we identified distinct GATA accumulation only on the X chromosomes of *B. huxleyi* (Fig. 1F) close to the region where the chiasma with the Y chromosome formed. However, it should also be noted that many of GATA arrays might be too small to be detected *via* the standard FISH method; therefore, the importance of GATA repeats in sex-chromosome expression regulation cannot be ruled out in the other two species that we studied.

Fluorescent *in situ* hybridization with a specific probe for $(TTAGG)_n$ arthropod telomeric repeats showed regular telomeric patterns in all three studied Zorapteran species. However, this does not specifically determine the Zorapteran telomeric motif, since FISH with $(TTAGGG)_8$ in *L. cacaoensis* showed the same telomeric pattern (Fig. S1C). Therefore, we suggest the Zorapteran telomeric motif is similar to the probe sequences, though, it needs to be determined in further studies. The TTAGG repeat is ancestral in insects and likely originated from widespread TTAGGG telomeric repeats (*Vítková et al., 2005*). Nevertheless, several insect lineages have lost the TTAGG telomeric motif; alternative motifs
have been detected in some of those lineages (reviewed in *Kuznetsova, Grozeva & Gokhman, 2020*). Interestingly, the ancestral TTAGG motif was not detected in two *Forficula* species (*Frydrychová et al., 2004*), which suggests that it is not present in at least some Dermaptera. In addition to the standard telomeric pattern, pericentric interstitial telomeric sequences (ITSs) have been found in the largest chromosome of *B. huxleyi* (Figs. 1A, 1C). Interestingly, size of this metacentric chromosome is approximately double the size of the largest autosomes in *B. kukalovae*, and some of these chromosomes also bear pericentric ITSs on their long arms. Paradoxically, ITSs are sometimes observed in chromosomal regions where Robertsonian fusions have occurred, even though one of the primary functions of telomeres is to prevent such rearrangements (*Murnane, 2012*). Therefore, it has been proposed that ITSs may originate from telomere inactivation followed by chromosome fusion within the telomere sequence (*Slijepcevic, 1998*) or by telomeric sequence insertion *via* telomerase during repairs of DNA double-strand breaks *via* the non-homologous end joining repair mechanism (*Nergadze et al., 2004*; *Nergadze et al., 2007*). Pericentric ITSs are distributed on both sides of centromeres in the largest chromosome of *B. huxelyi*; no signal is present directly in the primary constriction. Therefore, there is also the possibility that the ITSs were not generated during the process of putative fusion of two telocentrics but they might actually correspond to the ITSs that were already present in the telocentrics prior to fusion (*i.e.,* as observed in *B. kukalovae*). In that case, fusion would be facilitated by the centric regions of the telocentric chromosomes with a loss of proximal telomeres. Since we do not know the ancestral karyotype of *Brazilozoros*, the entire process could also be reversed: the ITSs on the telocentric chromosomes of *B. kukalovae* might instead be the products of chromosomal fission. Irrespective of directionality, both scenarios require that the pericentric ITSs are already present prior to fusion/fission. Conveniently, the spatial organization of telomeric repeats can be also driven by evolutionary mechanisms that generally act upon satellite DNA (*e.g.,* unequal and ectopic recombination, gene conversion and the formation of extra-chromosomal DNA circles) (reviewed in *Aksenova & Mirkin, 2019*).

## CONCLUSIONS

We have described the karyotypes of three Zorapteran species representing the subfamilies Latinozorinae and Spiralizorinae within the Spiralizoridae family. We determined chromosome numbers of 2n♂; ♀ = 42; 42 in *B. huxleyi* and 2n♂; ♀ = 44; 43 in *B. kukalovae*, 2n♂; ♀ =36; 36 in *L. cacaoensis*; the karyotypes are predominantly composed of telocentric chromosomes. None of the sampled species had holocentric chromosomes; *Kuznetsova, Nokkala & Shcherbakov (2002)* suggested as much for *U. hubbardi*. Interestingly, both X0 and XY SCSs were detected in Zoraptera; additional research is necessary to confirm which of these SCSs is ancestral in Zoraptera. Such work could help refine our understanding of SCS evolution in Polyneoptera and early branching Pterygota lineages. Fluorescent *in situ* hybridization using probes for histone H3 gene and $(CAG)_n$ microsatellites revealed one locus for each distributed in a stable pattern across all three of the studied species. That finding is indicative of the homeology of chromosomes bearing the loci. On the other

hand, the positions of the 18S rDNA cluster differed between the species in the *Brazilozoros* and *Latinozoros* genera, and the distribution of GATA repeat accumulations was unique in each of the species. Both $(TTAGG)_8$ and $(TTAGGG)_8$ probes visualized standard telomeric pattern in Zoraptera. Therefore, we suggest the Zorapteran telomeric motif is sequentially similar to the used probes but needs to be determined in further studies. Interstitial telomeric sequences have been detected in both *Brazilozoros* species; however, it is not clear whether those sequences were generated during chromosomal fusions or *via* the transposition mechanisms that can generally drive the distribution of repetitive DNA in genomes. This description of Zorapteran karyotypes serves as a stepping stone for further cytogenetic research focused on this group. Our results show that Zorapteran species may substantially differ at the karyotype and genomic organization level. Furthermore, both X0 and XY SCSs have been detected in Zoraptera. Ongoing research regarding evolution and variability of Zorapteran karyotype is critical for understanding the broad picture of karyotype evolution not only in Polyneoptera but also in the whole Insecta.

## ACKNOWLEDGEMENTS

We thank Ivan H. Tuf (Palacký University, Olomouc, Czech Republic) for his help and companionship during the field work in French Guiana. The authors are grateful to team of American Manuscript Editors for the grammatical review of the manuscript. We thank Marielle C. Schneider and one anonymous reviewer for valuable comments and suggestions that improved this paper.

### Funding

This research was supported by project GACR 22-05024S (Evolution of angel insects (Zoraptera): from fossils and comparative morphology to cytogenetics and transcriptomes) and by the Ministry of Education, Youth and Sports of the Czech Republic (SVV 260 686/2023) (M.J.). The funders had no role in study design, data collection and analysis, decision to publish, or preparation of the manuscript.

### Grant Disclosures

The following grant information was disclosed by the authors:
GACR: 22-05024S.
Ministry of Education, Youth and Sports of the Czech Republic: SVV 260 686/2023.

### Competing Interests

The authors declare there are no competing interests.

### Author Contributions

- Marek Jankásek conceived and designed the experiments, performed the experiments, analyzed the data, prepared figures and/or tables, authored or reviewed drafts of the article, collecting material, and approved the final draft.

- Petr Kočárek conceived and designed the experiments, performed the experiments, analyzed the data, authored or reviewed drafts of the article, collecting material, and approved the final draft.
- František Šťáhlavský conceived and designed the experiments, performed the experiments, analyzed the data, prepared figures and/or tables, authored or reviewed drafts of the article, and approved the final draft.

## Field Study Permissions

The following information was supplied relating to field study approvals (i.e., approving body and any reference numbers):

French Guiana (collecting place) as a part of European Union not required special permissions for EU citizens.

## DNA Deposition

The following information was supplied regarding the deposition of DNA sequences:

The sequences are available in the Supplementary File.

## Data Availability

The data are available in the figures, Materials and Methods and Results, and the Supplementary Files.

## Supplemental Information

Supplemental information for this article can be found online at http://dx.doi.org/10.7717/peerj.18051#supplemental-information.

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
