# Peer review of "Comparative cytogenetics of three Zoraptera species as a basis for understanding chromosomal evolution in Polyneoptera insects"

_PeerJ, doi:10.7717/peerj.18051_

## Round 0.1 · original submission · Major Revisions

Thank you for your manuscript. It was read by two skilled reviewers. While both saw it as a worthwhile contribution, there were some issues with reporting and methods that need to be clarified. Both reviewers require more detail in the methods and both requested information on species identification. Reviewer 1 was concerned about issues using FISH.

I'd like you to address all of the points raised by both reviewers. Further, as these could be substantial issues, I've decided to fla this as a major revision.

Thank you again.

**Language Note:** PeerJ staff have identified that the English language needs to be improved. When you prepare your next revision, please either (i) have a colleague who is proficient in English and familiar with the subject matter review your manuscript, or (ii) contact a professional editing service to review your manuscript. PeerJ can provide language editing services - you can contact us at [email protected] for pricing (be sure to provide your manuscript number and title). – PeerJ Staff

Reviewer 1 ·

Basic reporting

In this manuscript, the authors have performed comparative cytogenetic analyses in three Zorapteran species and provided detailed descriptions of the karyotype and sex chromosome system, including the distribution of some microsatellites, the histone gene, 18S rDNA and telomeric repeats. Overall, it seems that most parts of the manuscript are well written and most of the results are convincing. As little information on the karyotype and sex chromosome system of Zoraptera species is known so far, the results shown in this manuscript would help to understand the genome evolution, taxonomic and/or phylogenetic problems, etc. of this insect group.

I believe that this manuscript would be worthy of publication. However, I have some major concerns about the results and descriptions (see Experimental design). Therefore, several major revisions should be accomplished by the authors.

Experimental design

Major comments

(1) The taxonomic identification of the three Zoraptera species used in this manuscript
L100-L107: How did the authors make the taxonomic determination of the three Zoraptera species? Did the authors use some morphological characters or DNA barcoding etc. to identify the species? I think this information is very important. Please describe them.

(2) Telomeric repeats
The authors claim that in the three Zoraptera species the telomeric sequences consist of a (TTAGG)n telomeric motif that is considered to be an ancestral motif of telomeres not only in insects but also in arthropods. This is a very interesting finding. However, I have major concerns about the FISH protocol, especially the washing of the slides after hybridization that the authors used in this study. According to the descriptions in the methods (L117-L128), the authors seem to use low stringency washing after hybridization for this telomere-FISH compared to FISH with 18S rDNA and histone gene probes. If the washing is done with low stringency, similar sequences (e.g., TTAGGG etc.) could be hybridized to chromosomes. To avoid this and to know the actual telomeric motif in the three species, I think the authors should show (or at least describe) the results of negative control, such as mapping of other telomeric motifs, in this manuscript.

(3) The descriptions in results

3-1. L178-L179, “20 homomorphic”: Fig. 1D clearly shows 21 bivalents (20 autosomal pairs and one sex chromosome pair). In Fig. 1E, on the other hand, it looks to me like 22 bivalents (21 autosomal pairs and one sex chromosome pair). How can the authors explain the differences in the number of chromosomes?

3-2. L200-L203, “the sentences for metaphases II”: I struggled to understand why the authors show chromosomes in metaphase II and what the authors want to claim with these sentences. What kind of information do the authors want to present with the chromosomes in metaphase II and these sentences? Do the authors want to show differences in chromosome number between individuals with these sentences? In my opinion, Fig. 2A and B clearly show that the males of this species have a karyotype of 2n=43 with an X0 sex chromosome constitution, and it would not make sense to show chromosomes in metaphase II and the corresponding descriptions. I would therefore suggest deleting these sentences.

3-3. L202-L203, “heteropycnotic X chromosome (Fig. S1C)”: According to the legend of Figure S1, Fig. S1C should show male Latinozoros cacaoensis (L830). Why did the authors cite Fig. S1C in the section of Brazilozoros kukalovae?

(4) Discussion

4-1. L340-L342, “the cluster was shifted slightly more distally from the centromere.”: In the figure showing the results of microsatellite (CAG)n mapping, the authors show mitotic chromosomes (Fig. 2) in one species and meiotic chromosomes (Figs. 1 and 3) in other species. Chromosome condensations and structure should be different for mitotic and meiotic chromosomes, which could lead to the different mapping positions. Therefore, I think it is difficult to claim this, at least based on the results in this manuscript.

4-2. L339-L376: I think that the descriptions in these paragraphs are not relevant to the results shown in this manuscript. These are too much of a hypothesis and speculation. I would therefore suggest either rephrasing them to a simpler description or deleting them.


Minor comments

L131, “The 18S rDNA probe”: Is this 18S rDNA a fragment? If so, “The fragment of 18S rDNA” would be better.

Supplemental file S2: Is there any accession numbers for these genes?

In the entire manuscript: The authors appear to be using 18S rRNA in some places but 18S rDNA in others. Is there a reason for these differences?

L296-L300: The content in this sentence is already described in results. It would be better to delete this sentence.

L811, “DAPI (blue) (B, D-E)”: F is missing.

Validity of the findings

no comment

·

Basic reporting

This paper has a well-substantiated theoretical framework and describes the chromosomes of three species of Zoroptera, comparing the data with other phylogenetically related groups of insects. The Zoraptera are particularly interesting for carrying out this type of study, considering the scarcity of karyotype information and its phylogenetical position within the Insecta. The results have excellent quality and are in general way well documented. Therefore, I recommend this paper for publication after minor revision.
Some specific comments to the authors are below mentioned:
Introduction
This section is embracing with relevant bibliography. However, I suggest includes more detail about the inner classification of Zoraptera, such as families and subfamilies.

Lines 72-73: “ we provide here the first cytogenetic analysis in Zoraptera in form of karyotype descriptions” This sentence could be changed. The present paper is not the first cytogenetic description of Zoraptera. There are data of chromosome number to Usazoros hubbardi, despite the results did not have a good quality.

Material and Methods
The methodology needs more details. I suggested includes the local the vouchers specimen deposition. The number the cells measured to determine the morphological classification of chromosomes and the relative chromosome length also be specified.

Results
Figures – The images have excellent quality, but I suggest includes in the figures 1 and 3, the X and Y chromosomes obtained in mitotic metaphase of males if it is possible.
Lines 186 - 188 – Please, revise the description of H3 signals – are this gene located in pericentromeric or terminal chromosome region?


Discussion
Lines 249-250 “However, the karyotype evolution modes of other Polyneoptera orders also remain unknown due to a lack of knowledge about their karyotypes” – Please, rewrite this sentence.

Lines 253 – 258 – I suggested that you improve the discussion about the holocentricity in
Usazoros hubbardi mentioning the family in which this species is included. This point should be relevant because in some arachnid groups the holocentric chromosomes seem to be restricted to only some families.

Lines 270-271 – “Evidence for such fusions is the presence of large biarmed chromosomes in species with reduced 2n (potentially B. huxleyi and L. cacaoensis)”. Your most important issue of comparison here are the species of the same genus - Brazilozoros huxleyi with 2n♂=42=40+XY and Brazilozoros kukalovae with 2n♂=43=42+X0.

Lines 294-321 – The discussion about the location of rDNA sites is very long and not focused on the results obtained. Additionally, some information are repetitive, e.g. lines 300-301 -”rDNA clusters are most frequently found on short chromosome arms in animals and plants” and lines 309-310 – “is the least common type of rDNA localization in animals and plants”. Please, revise this point of discussion.

Lines 350-376 – The discussion on GATA clusters could be shorted, focusing in the comparison on related groups and the data obtained in the present paper.

Experimental design

All the comments were included in the "Basic reporting".

Validity of the findings

All the comments were included in the "Basic reporting".

---

## Round 0.2 · Minor Revisions

Thank you very much for your revision and your patience in waiting for this response.

I can see you handled most of the comments suggested by two reviewers. One reviewer had some minor issues raised in the round that will need addressed.

Please handle these and I look forward to seeing a revision.

Reviewer 1 ·

Basic reporting

In this revised manuscript, the authors have addressed all comments from the reviewers, and the revised manuscript has been significantly improved. I have only a few comments on this revised manuscript.

Regarding telomere-FISH, I appreciate that the authors have also shown FISH results with (TTAGGG)n probes. As I said in the previous review, my major concern was that the authors used low-stringency washing method after hybridization for this telomere-FISH compared to FISH with 18S rDNA probe. As the authors show in this revised manuscript, the (TTAGGG)n probe actually seems to hybridize to the telomeric region when washed with low-stringency. Have the authors ever used the high-stringency washing methods after hybridization such as that shown in L176 (Sahara et al. 1999) for telomere-FISH in this species? Please consider using high-stringency washing methods after hybridization.

L313: Please delete “Clusters of”.

L781, “white arrowheads”: Please delete “white” as there is a black arrowhead in Fig. 1E.

Experimental design

no comment

Validity of the findings

no comment

·

Basic reporting

All the suggestions were considered by the authors in this revised version of the manuscript, i.e. information were included in Introduction and Material and Methods sections, and the Discussion was reduced, emphasizing the data obtained in the present work. In have no additional suggestions on the text, and I think that the current version of the manuscript can be accepted for publication.

Experimental design

No comment

Validity of the findings

No comment

---

## Round 0.3 · accepted · Accept

Thank you for your efforts on this paper. The reviewers and I agree that you have accomplished all their requests and your paper is acceptable.

Thank you again for your submission.

Reviewer 1 ·

Basic reporting

I appreciate that the authors have conducted further experiments to verify the telomeric motif in these species and have presented new results. According to the results presented by the authors in the revised manuscript, the telomeric motifs in these species become more and more interesting to me. Anyway, the revised manuscript looks well done and author’s responses make sense. Therefore, I have no comments this time, and I believe that the revised manuscript can be accepted.

Experimental design

no comment

Validity of the findings

no comment